# Comparative Failure Study of Different Bonded Basalt Fiber-Reinforced Polymer (BFRP)-AL Joints in a Humid and Hot Environment

**DOI:** 10.3390/polym13162593

**Published:** 2021-08-05

**Authors:** Yisa Fan, Jinzhan Guo, Xiaopeng Wang, Yu Xia, Peng Han, Linjian Shangguan, Mingyue Zhang

**Affiliations:** 1School of Mechanical Engineering, North China University of Water Resources and Electric Power, Zhengzhou 450000, China; fanyisa123@163.com (Y.F.); guruci@163.com (J.G.); 15038405185@163.com (X.W.); zxc18437350069@163.com (Y.X.); 2College of Tobacco Science, Henan Agricultural University, Zhengzhou 450002, China

**Keywords:** BFRP, adhesive-bonded joint, mechanical property, high temperature, Fickian law

## Abstract

Fiber-reinforced polymer (FRP) materials are increasingly used in automotive industrial fields to achieve lightweight. In order to study the influence of high temperature and high humidity on the bonding structure between different materials, this paper selects basalt fiber-reinforced resin composite materials (BFRP) and aluminum alloy (Al), and uses Araldite® 2012 and Araldite® 2014, two adhesives, to make single lap joints (SLJs). The aging test was carried out for 0 (unaged), 10, 20, and 30 days under the environment of 80 °C/95% relative humidity (RH) and 80 °C/pure water. In this work, simple Fickian law was used to simulate the hygroscopic change law of dumbbell specimens of two adhesives and BFRP in a pure water environment. It was discovered that Araldite® 2012 is most affected by moisture, but the time to reach the maximum water absorption in Araldite® 2014 was shorter than in Araldite® 2012. The failure strength of the joint was obtained through a quasi-static tensile experiment, and it was found that the Araldite® 2014 adhesive joint first increased and then decreased in a high temperature environment. The strength increased by 11.63% after 20 days of aging under an 80 °C/95%RH environment, and increased by 16.66% after 10 days of aging under an 80 °C/pure water environment, which indicates that post-curing reaction occurred. The strength of Araldite® 2012 joints showed a downward trend. After 30 days of aging, it reduced by 40.38% under an 80 °C/95%RH environment and 41.11% under an 80 °C/pure water environment. By observing the load-displacement curve, it was found that, as time increased, the slope of the curve decreased, indicating that the stiffness of the bonded joint decreased with time. The failure modes of the joints were analyzed by macroscopic images and microscopic SEM methods, and the results showed that the surface failure transitions from a mixed failure to a complete tear failure over time. The failure of the basalt fiber/resin interface was because the interaction between the epoxy resin in the adhesive and the epoxy resin in BFRP was greater than the force between the basalt fiber layer and the epoxy resin layer in the BFRP sheet.

## 1. Introduction

In recent years, with the increasingly prominent energy shortage, the development of new energy vehicles is becoming one of the main solutions to the problem, but endurance capacity has always restricted the development of new energy vehicles. At present, the replacement of lightweight materials is becoming the mainstream. Further, composite materials are increasingly used as lightweight materials in vehicles. Among many composite materials, basalt fiber-reinforced resin composite materials have the advantages of high specific strength, high specific modulus, corrosion resistance, high temperature resistance, good thermal stability, and good impact resistance [1]. Compared with carbon fiber-reinforced resin (CFRP) composite materials, BFRP composite materials have the advantages of a wide range of raw materials and low cost. Compared with glass fiber-reinforced resin (GFRP) composite materials, it has the advantages of low moisture absorption and environmental protection, and has a wide range of applications [2]. However, the connection technology of FRP and other materials restricts the further development of the application of this material. Traditional connection methods, such as riveting, bolt connection, threaded connection [3], and so on, are not suitable for FRP connection. These methods usually result in adhesion damage, fiber tearing, and unavoidable stress concentration. Therefore, the joint will fail at a lower level than expected [4]. As a new type of structural connection technology, bonding technology can ensure that different materials achieve sufficient strength without destroying the adhesive, and it has the advantages of uniform stress distribution, good fatigue performance, and light weight [5]. Therefore, the use of bonding technology to connect dissimilar materials (metal and non-metal) is playing an increasingly important role in the automotive industry.

The service environment of the car is roughly between −40 °C and 80 °C, of which the 80 °C high temperature and high humidity environment has a significant impact on the bonding performance, especially the mechanical properties and fatigue resistance of the composite material will have some adverse effects [6]. In terms of temperature research, Avendaño et al. [7] selected an acrylic adhesive combined with biopolymers in single lap joints (SLJs), and conducted joint experiments at −30, 23, and 80 °C. The results showed that, owing to the softening of the adhesive, the strength of the joint at high temperatures decreased significantly. Yao et al. [8] analyzed the effect of temperature on BFRP-steel SLJs, and they found that the average bond strength increased in the range of −25–50 °C, but decreased significantly in the range of 50–100 °C, and shear stiffness increased in the range of −25–25 °C, but decreased in the range of 25–100 °C. Therefore, it was concluded that high temperature had a significant effect on mechanical properties. Ke Lu et al. [9] studied the bond strength degradation mechanism of two bonded joints at high temperatures. As the temperature increased, the strength of Araldite® 2014 joints decreased mainly owing to the reduction in the adhesion of the bonding layer to the CFRP interface, while the decrease in J133 adhesive was mainly due to the reduction in the degree of adhesion between the bonding layer and the steel. Chiara Borsellino et al. [10] used double lap joints (DLJs) for testing. Different failure modes were observed at room temperature and high temperature. At room temperature, they were “adhesive failure”, “adhesive failure”, “light fiber tear failure”, or “mixed failure”, while only “adhesive failure” was observed at high temperature.

On the other hand, in the research on the effect of moisture on the bonding structure, when the moisture diffused into the adhesive layer, it filled the free volume in the form of free water or bound water, thereby inducing the plasticization of the adhesive and reducing the glass transition temperature (*T*_g_) [11]. Zheng et al. [12] studied the degree of fit between the simple Fickian model and the sequential dual Fickian (SDF) model in terms of water diffusion, and found that the sequential dual Fickian model for toughened epoxy adhesives can better describe the moisture absorption behavior of the adhesive. Florian Cavodeau et al. [13] studied the influence of water diffusion mechanism on the durability and adhesion of “metal-adhesive-metal” bonded specimens at 40 °C. It was found that the hygroscopicity and volume increase are reversible, and the durability of glued joints mainly depended on the diffusion kinetics of water and the degradation of the interface, and the diffusion of water to the interface was related to the uniformity of contact between the adhesive and the metal matrix. Han et al. [14] studied the slight degradation of FM1000 and FM 73 bonded joints in water and simulated the diffusion process of water. Ahmad Reza Zaeri et al. [15] found that the relative mass absorption of Araldite® 2015 in equilibrium was higher than that of Araldite® AV 138 under the same environmental conditions. They believed that Araldite® 2015 was more susceptible to moisture than Araldite® AV 138. Dhakal et al. [16] used nanoindentation and bending tests. Through two different samples, dry and wet, the effect of the water absorption rate of fabric, flax, and jute fiber-reinforced bio-resin matrix composites on the elastic modulus and strength was studied. It was found that the strength and elastic modulus of the composites were easily affected by moisture absorption.

From the above research, it is found that temperature and humidity have different effects on the bonding performance of the joint, but the influence of the two factors may not simply be superimposed. In particular, humidity and water immersion under high temperature conditions not only affect the failure load of the joint, but the failure mechanism may also change. Based on the service environment of the vehicle during use, it is necessary to carry out research on the high humidity and water immersion environment of the bonding structure. In terms of the combined effect of damp-heat aging, Stazi et al. [17] used different types of adhesives to connect GFRP, and characterized their behaviors after damp-heat degradation and ultraviolet radiation. It was found that environmental exposure had little effect on the failure load of the joint, but the elongation increased significantly and the stiffness decreased. The moisture absorption characteristics could explain the non-monotonic changes in the mechanical properties of the adhesive. Barbosa et al. [18] studied the effect of water absorption on the mechanical and physical and chemical properties of brittle epoxy resins at different temperatures below *T*_g_. Increasing the temperature accelerated the diffusion rate of water molecules, and as the moisture absorption increases, the degradation of mechanical properties tends to be significant. Zhang et al. [19] studied the mechanical behavior of bonding joints between aluminum and steel substrates under high temperature and high humidity, and the results showed that high temperature and high humidity significantly affected the bonding strength. Rais et al. [20] studied the impact of hydrothermal aging on the impact strength. After soaking in distilled water at 40 °C for 8 weeks, the impact performance of the experimental sample was reduced by about 40% compared with the control sample. The coupling effect of water and temperature (hydrothermal effect) is usually more harmful than the influence of a single factor [21]. David et al. [22] used an experimental method to study the bonding performance of the joint under the coupling effect of damp heat and tension, and found that, under the combined effect of damp heat and tensile load, the failure strength of the joint was greatly reduced.

This article is devoted to studying the influence of temperature and humidity coupling on the mechanical properties of BFRP-Al bonded joints. Two adhesives with different properties were selected, namely Araldite® 2012, a tough epoxy adhesive, and Araldite® 2014, a brittle epoxy adhesive. The research used 80 °C/95%RH and 80 °C/pure water as the aging environment, and conducted aging experiments for 10 days, 20 days, and 30 days, respectively; tested the mechanical changes of the joints at different aging times; analyzed the average failure strength of the joints and the change laws of the load–displacement curve; and characterized the failure morphology of the joints at different aging times. At the same time, the dumbbell-shaped specimens and the BFRP substrates were tested for water absorption, and the water absorption conditions of the materials were studied to provide help for the analysis of the failure mechanism of the bonded joint.

## 2. Materials and Methods

### 2.1. Materials

The base material of BFRP-Al joint was made of BFRP composite plate and 5052 Al. The thickness of the BFRP composite sheet was 2 mm, and it was made of twill, unidirectional prepreg, BFRP was bought from Zhongdao Technology Co., Ltd. (Jilin, China). The fiber layer was laid in [0/45/0/45/0/45] s, and the density was 600 g/cm^2^. 5052 Al is a high-quality aluminum alloy produced by heat treatment and the pre-stretching process, with a thickness of 2 mm. The specific parameters were provided by the manufacturer, as shown in Table 1 and Table 2. Araldite® 2012 and Araldite® 2014 were adopted as adhesives, and was provided by Huntsman Advanced Materials Co., Ltd. (Shanghai, China). Araldite® 2012 is a high-viscosity liquid adhesive with high strength, good toughness, and room temperature curing. Araldite® 2014 is a room temperature curing brittle epoxy adhesive with thixotropy and environmental corrosion resistance. The specific parameters of Araldite® 2012 and Araldite® 2014 are shown in Table 3.

### 2.2. Single Lap Joint Design

In this paper, SLJs were selected to study the aging law of BFRP-Al joints. The specific dimensions of the specimens were shown in Figure 1. The lap length and width were both 25 mm, and the thickness of the adhesive layer was 0.1 mm. The production standard of bonded joints referred to ISO4587: 2003 [23].

In order to prevent other variables from affecting the experiment, all test pieces adopt a uniform and standard bonding process, and were carried out under dust-free, room temperature 25 ± 3 °C, and relative humidity 50 ± 5%. The surface treatment of 5052 Al substrate was firstly polished with 80# sandpaper to remove the oxide layer and increase the roughness. Then, the following steps are carried out: use acetone to wipe the bonding surface of the aluminum alloy and BFRP composite material to remove dust and grease. Wait 15–20 min after wiping. After drying, use a two-component (Araldite® 2012: curing agent = 1:1, Araldite® 2014: curing agent = 1:1) glue gun for sizing. Then, use a gasket 0.1 mm thicker than the substrate to control the thickness of the adhesive layer, and complete the bonding of the test piece in a special fixture. The bonding fixture is shown in Figure 2. There is a layer of polytetrafluoroethylene on the surface of the fixture to prevent excess adhesive from bonding the joint and the fixture. After the bonding is completed, the test piece is removed from the fixture to cure at room temperature for 120 h, and the remaining glue is cut off after curing. Then, put the prepared test piece into the high and low temperature damp heat alternating experiment box (Weiss Equipment Experiment Company WS-1000), and complet the 80 °C/95%RH and 80 °C/pure water aging experiment.

### 2.3. Specific Test Methods

#### 2.3.1. Experimental Design

In order to study the aging behavior of BFRP-Al joints under 80 °C/95%RH and 80 °C/pure water environment, 0 (unaged), 10, 20, and 30 days were selected as the aging time points. Each environment was divided into four groups according to the four experimental times, a total of eight groups, and each group selected five test pieces. After the specimens were cured, the specimens were put into the high-low-temperature damp-heat alternating experiment box, and the aging of the corresponding environment and time was carried out. After reaching the corresponding time, it was taken out and the remaining strength of the bonded joint was tested after placing it for 8 h.

#### 2.3.2. Water Absorption Test

Considering that the bonding body was the main factor of the bonding strength of the joint, in order to better study the change law of the bonding agent under the temperature and humidity conditions, this paper designed a set of dumbbell-shaped specimen standard molds, as shown in Figure 3. With reference to ISO 527-2-2012 [24], a dumbbell-shaped test piece with a thickness of 2 mm was produced. The specific size and shape are shown in Figure 4 to study the moisture absorption performance of the adhesive body. A high-precision analytical balance was used to test once every 24 h. At least three samples in each group were used to record the quality and original quality of the dumbbell-shaped specimens made of the two adhesives and BFRP at different times. In the process of weighing the test piece, disposable dust-free rubber gloves were first put on, then the test piece was taken out of the high and low temperature experiment box, and then it was gently wiped with absorbent paper to remove surface moisture. Finally, it was measured on a high-precision analytical balance. The balance accuracy was 0.0001 g. The whole process was fast and needed to avoid the influence of other factors.

The calculation formula of water absorption is as follows:(1)$$M_{t}=\frac{W_{t}-W_{0}}{W_{0}}\times 100\%$$
where $W_{t}$ represents the quality at time *t*, and $W_{0}$ represents the original quality.

#### 2.3.3. Differential Scanning Calorimetry

For adhesive materials, glass transition temperature (*T*_g_) is one of the important parameters that determine its use environment. In order to study the influence of the aging environment on the material *T*_g_, a differential scanning calorimeter (Mettler Toledo, DSC 3+, Switzerland) was used to analyze the adhesive before and after aging. The mass of the adhesive test sample was about 5 mg, and the adhesive samples come from the failure section of the bonded joints. The DSC test was carried out in a nitrogen environment, the heating/cooling rate was 5 °C/min, the temperature range was −70–200 °C, and each set of tests was repeated three times. The heating process was repeated twice for each test, the first time to remove the influence of the thermal history and the second heating process was taken as the final result.

#### 2.3.4. Quasi-Static Tensile Test of Single Lap Joint

Xinguang Universal Testing Machine (Jinan Xinguang Testing Machine Manufacturing Co., Ltd., Jinan City, China) was used to conduct tensile testing of BFRP-Al single lap joints treated with different environments and aging times. A 2 mm thick gasket was clamped at both ends of the test piece to eliminate the bending stress existing in the stretching process. The joint was stretched by the testing machine at a constant speed of 2 mm/min until it was broken. During the stretching process, the force-displacement curve was recorded by a computer system connected to the universal testing machine. The quasi-static stretching is shown in Figure 5.

## 3. Results

### 3.1. Water Absorption

Generally speaking, Fickian second law [25] is used to describe the unsteady diffusion process, which is usually called simple Fickian Law. The derivation process is as follows.

Assuming that the mass is conserved in the process of molecular re-diffusion, molecules with flux enter the space and molecules with flux leave within unit time, as shown in Figure 6. Then, in unit time, the change of the concentration in the cuboid is as follows:(2)$$\frac{C\left(t+\tau \right)-C\left(t\right)}{\tau }=-\frac{J\left(x\right)-J\left(x+2h\right)}{2h}$$

*C*(*t*) represents the water concentration at time *t* and *C*(*t* + *τ*) represents the water concentration at time *t* + *τ*.

When *τ*→0, 2*h*→0, Equation (2) is as follows:(3)$$\frac{\partial C\left(x\right)}{\partial t}=-\frac{\partial J\left(x\right)}{\partial x}$$

*C*(*x*) represents the water concentration at the spatial coordinate *x*.

Substituting Fickian first law $J\left(x\right)=-D\frac{\partial C\left(x\right)}{\partial x}$ into Equation (3), we can obtain the following:(4)$$\frac{\partial C\left(x,t\right)}{\partial t}=D\frac{\partial C^{2}\left(x,t\right)}{\partial x^{2}}$$

*C*(*x*,*t*) represents the water concentration at the space coordinate *x* and time *t* and *D* is the diffusion coefficient.

Equation (4) is the form of Fickian second law, which is a parabolic partial differential equation. Equation (4) is combined with the boundary conditions and the initial conditions of the flat sheet with an absorption thickness of 2*h*, as shown in (5):(5)$$\left\{\begin{array}{c} \frac{\partial C\left(x,t\right)}{\partial t}=D\frac{\partial C^{2}\left(x,t\right)}{\partial x^{2}}\\ C\left(x=\pm h,t\right)=C_{\infty },C\left(x,t\leq 0\right)=0 \end{array}\right.$$

The solution of Equation (5) is as follows:(6)$$\frac{C\left(x,t\right)}{C_{\infty }}=1-\frac{4}{\pi }\overset{\infty }{\underset{n}{\sum }}\frac{{\left(-1\right)}^{n}}{2n+1}\exp\left(\frac{-D{\left(2n+1\right)}^{2}\pi^{2}t^{2}}{4h^{2}}\right)\cos\frac{\left(2n+1\right)\pi x}{2h}$$

Integrating the spatial variable x of Equation (6), the following can be obtained:(7)$$\frac{M_{t}}{M_{\infty }}=1-\frac{8}{\pi^{2}}\overset{\infty }{\underset{n=0}{\sum }}\frac{1}{{\left(2n+1\right)}^{2}}\exp\left(\frac{-D{\left(2n+1\right)}^{2}\pi^{2}t}{4h^{2}}\right)$$

*M_t_* represents the saturated mass absorption.

The diffusion coefficient *D* is calculated using Formula (8), and the calculation results are shown in Table 4:(8)$$\frac{M_{t}}{M_{\infty }}=\frac{2}{h}\sqrt{\frac{Dt}{\pi }}$$

MATLAB software was used to fit the curve of Fickian law to the experimental data, as shown in Figure 7. And their main data of moisture uptake are shown in Table 4. It can be seen from Figure 7 that the curves based on the simple Fickian law initially grew relatively fast, and then gradually slowed down and reached saturation. The basic trend was consistent with the experimental data and can be roughly divided into three regions. In the first few points of the first stage, it was found that the experimental data and the fitted data in the two figures had a large gap, but the gap between the two also showed different performance. Although the gap between Araldite® 2012 dumbbell-shaped specimens was large, but the time was short, Araldite® 2014 was small, but longer comparatively. The reason for this may be the big jump of the sample from 50 ± 5% of the air humidity to 100% of the pure water humidity, while the fitted data showed a gradual increase in the humidity, so the experimental data are slightly larger than the fitted data. At the same time, we can also saw that the diffusion coefficient of Araldite® 2014 was greater than that of Araldite® 2012, which was mainly because its maximum water absorption was smaller, thus the speed to reach the saturation value was faster and the time was shorter.

In the water diffusion of Araldite® 2012 dumbbell-shaped specimens, the first stage should be based on filling of pores and cavities with water, and at the same time, supplemented by the reaction of adsorbed water molecules with some hydrophilic functional groups, both of which occurred simultaneously [26]. The second stage was when the water molecules in the pores and cavities reached saturation. At this time, the reaction between water molecules and some hydrophilic groups was the main reason. As a polar molecule, water formed hydrogen bonds with hydroxyl groups, so the hydrogen bonds between chains can be broken, thereby increasing the length of the hydrogen bonds between segments [27] and weakening the stress between polymer molecular chains. The third stage was the saturation of water absorption. In fact, it can be clearly observed that the experimental measurement data of the second stage were quite different from the simple Fickian Law simulation data. This was mainly because of the limitations of simple Fickian law simulation: simple Fickian law only assumes that water diffuses into the material and stays in the free volume, while ignoring the reaction of water molecules with certain functional groups in the material [28]. In practice, when the free volume was filled with water molecules, the reaction with the functional groups was to push the water molecules to absorb more power. In the simulation of simple Fickian law, this factor is ignored, meaning the increase rate of water absorption can only gradually decrease.

In Araldite® 2014 adhesive joints, the general trend shown in Figure 7b was similar to that in Figure 7a, but the maximum water absorption rate was nearly half the size of that of Araldite® 2012 dumbbell-shaped specimens. In the second stage, the rate of decrease of water absorption was faster than the simulation of simple Fickian law, that is, the experimental curve was below the simulation curve of simple Fickian law.

Figure 7c shows the water absorption of BFRP in pure water and a simple Fickian law simulation. In Table 4, we can see that the diffusion coefficient and maximum water absorption of BFRP were relatively small. Combined with Figure 7c, it was found that the experimental data in the first stage were mostly lower than the fitted data. This seemed to be very different from the previous two adhesives. In fact, the reason for it was because of the presence of basalt fibers. Basalt fibers in the BFRP used in this experiment accounted for 65–70%. Because it has the characteristic of not absorbing water or absorbing little water, and the simple Fickian law simulation assumes that all substances absorb water, so the water absorption rate in the first stage was low, and the maximum water absorption rate and diffusion coefficient were also small. Dhakal et al. [16], mentioned three mechanisms of water diffusion to fiber-reinforced materials:The first involves diffusion of water molecules inside the micro gaps between polymer chains.The second involves capillary transport into the gaps and flaws at the interfaces between fiber and the matrix.The third involves transport of micro-cracks in the matrix arising from the swelling of fibers (particularly in the case of natural fiber composites such as flax and jute).

In BFRP, the phenomenon of the third type of fiber expansion should be no or less, but the epoxy resin between different layers of basalt fiber cloth formed internal stress due to the different expansion volume, and the existence of these internal stresses can also cause micro-cracks. Finally, when the water absorption rate reached saturation, the combined water and free water in the composite material would cause fiber softening and weakening of the adhesion of the fiber matrix, which would reduce the material properties [16].

Based on the above analysis, the following conclusions can be drawn: Araldite® 2012 was most affected by moisture, but the time for Araldite® 2014 to reach the maximum water absorption was shorter than that for Araldite® 2012, and the least affected by moisture was BFRP.

### 3.2. DSC Analysis

DSC equipment was used to analyze Araldite® 2012 and Araldite® 2014 adhesives that were not aged and aged at 80 °C/95%RH and 80 °C/pure water for 30 days, as shown in Figure 8.

It can be seen in Figure 8 that the glass transition temperature after aging was lower than that of the unaged glass in the four cases. At 80 °C/95%RH, Araldite® 2012 decreased by 10.51% and Araldite® 2014 decreased by 20.69%. In an 80 °C/pure water environment, Araldite® 2012 decreased by 2.94% and Araldite® 2014 decreased by 8.77%. Generally speaking, this meant that the molecular chain of the adhesive is broken under the environment of high temperature and high humidity. As a polar molecule, water formed a hydrogen bond with the hydroxyl group. Therefore, the hydrogen bond between the chains can be broken, thereby increasing the length of the hydrogen bond between segments [27]. Such results actually indicated that the aging environments weakened the stress between polymer molecular chains, increased the mobility of polymer molecular chains, and reduced the crystallinity of polymer molecular chains. They were also the reason for the decrease in *T*_g_ and the root cause of microstructure damage such as fiber debonding and matrix cracking [29]. Comparing the above data, it can be found that, in Araldite® 2012 and Araldite® 2014 adhesives, the *T*_g_ of the pure water environment was lower than that in the 95%RH environment. This was because of the fact that the order of water molecules inside the adhesive was restricted by external water molecules in a pure water environment, which made the system reach a balance [30]. As time goes by, the system will gradually become more balanced, so its degree of change was smaller than that of a 95%RH environment.

### 3.3. Joint Failure Strength Analysis

Through the quasi-static tensile test, the mechanical properties of the joints treated in an 80 °C/95%RH and 80 °C/pure water environment were tested. The obtained mechanical property data were statistically processed, and the average failure strength of the joint was obtained with the change rule of the aging time, as shown in Figure 9.

Observing Figure 9a, it can be seen that, for Araldite® 2012 joints, under the environment of 80 °C/95%RH, the average failure strength after 10, 20, and 30 days of aging decreased by 32.30%, 38.09%, and 40.38%, respectively. Under the environment of 80 °C/pure water, the reduction was 33.70%, 39.32%, and 41.11% after 10, 20, and 30 days, respectively. It can be seen that the curves under the two environments were very similar, mainly because the 80 °C/95%RH environment was almost the same as the 80 °C/pure water environment. The curve declined faster in the aging time of 0 to 10 days, and declined more slowly in the aging time of 10 to 30 days. In addition, it can be observed that the average maximum failure strength for 10 days and 20 days under the 80 °C/95%RH environment was slightly greater than in the 80 °C/pure water environment, while it was significantly less than the 80 °C/pure water environment at 30 days. Generally speaking, with the extension of the aging time, the maximum failure strength of the joints continued to decrease, and the degree of aging continued to deepen.

It can be obtained from Figure 9b that, for Araldite® 2014 joints, under the environment of 80 °C/95%RH, the average failure strength increased by 3.70% after 10 days of aging, increased by 11.63% in 20 days, and decreased by 2.18% in 30 days. In an environment of 80 °C/pure water, the average failure strength increased by 16.66% after 10 days of aging, decreased by 10.78% in 20 days, and decreased by 12.50% in 30 days. Among them, it can be found that the failure strength of the Araldite® 2014 joints under the 80 °C/95%RH environment reached the peak at 10 days, and the failure strength of the Araldite® 2014 joints under the 80 °C/pure water environment reached the peak at 20 days. Therefore, it can be inferred that the curing speed after the initial stage was greater than the aging corrosion speed under a high temperature and high humidity environment. This fact was mainly related to two points. One was that the high temperature led to a faster post-curing [31]; the other was that water molecules cannot quickly enter the adhesive and were in the process of gradually opening the bonding line [32]. In the later period, the aging effect began to become prominent and gradually occupied the main position. The water molecules completely opened the bonding line to enter the pores and cracks, first deposited in the pores and cracks, and then increased the pores and cracks under the continuous movement and action of water molecules [32]. The high temperature of 80 °C promoted the irregular movement of water molecules and accelerated the emergence of this situation. When the water molecules were gradually saturated in the pores and cracks, the water molecules will produce some chemical effects in the adhesive and the surface of the metal substrate. Inside the adhesive, it was mainly manifested by the formation of hydrogen bonds by water molecules and the breaking of polymer chain segments and plasticization [29], while on the surface of metal substrates, it was mainly hydration [33].

Figure 10 shows the comparison of two adhesive joints in the same environment. From Figure 10b in the pure water environment, it can be found that the average maximum failure strength of Araldite® 2012 joints in the late aging period (20–30 days) only changed by 3.46%, and that of Araldite® 2014 joints only changed by 2.07%. The changes between the two kinds of joints were very small, and they seemed to tend to a stable state. In Figure 10a, in the 80 °C/95%RH environment, the average maximum failure strength of the Araldite® 2012 joint had a change of 13.39%, and that of the Araldite® 2014 joint had a change of 12.40%. In comparison, the strength of the joints in the 80 °C/95%RH environment had dropped more and more obviously (13.39% and 12.40%), indicating the continuous effect of aging. In the 80 °C/pure water environment, the drop was very small (3.46% and 2.07%). The situation was also because of the fact that the order of the water molecules inside the adhesive in the pure water environment was restricted by the external water molecules, which meant the system reached a balance [34]. Over time, the changes in the system will gradually become smaller. In 80 °C/95%RH, although it was a high humidity environment, the external water molecules were not enough to restrict the internal water molecules of the adhesive. In this way, a conclusion consistent with the results of the DSC curve was reached.

### 3.4. Failure Displacement Analysis

Through the quasi-static tensile test, a computer connected to the tensile machine was used to record the load–displacement curves of different joints, as shown in Figure 11.

Observing Figure 11a,c, it can be found that the unaged curve was very different from the aging curves, which was mainly manifested in the maximum failure strength and tensile displacement. In Figure 11d, there was a significant difference between the aging for 10 days and the other curves. The reason was still post-curing [31], which increased the degree of cross-linking of the polymer [35] and hardened the epoxy resin. It can be clearly seen from Figure 11a that the slope of the curve continues to decrease as the aging time increases, and the degree of decline increased with the increase in the number of aging days, which indicated that the joint stiffness decreased with the increase in aging time. A similar trend was basically shown in Figure 11c, which was caused by the weak cohesion between the substrate and the bonding interface [33] owing to the aging environment. Comparing the two pictures in Figure 11a,b, it can be found that the curves of Araldite® 2012 joints were a little smoother, while the overall Araldite® 2014 joints showed a bilinear trend. This was because the properties of the two adhesives were different. Araldite® 2014 was a brittle adhesive. The turning point of bi-linearity was the generation of microcracks, which will significantly affect the stiffness [36]. During the stretching process of the Araldite® 2014 adhesive joint, a very subtle fracture sound will be heard at the turning point of the bilinear, and there will be a very crisp fracture sound when all the joints were broken. Careful observation of the four pictures in Figure 11 showed that, when the displacement is small, the overlap of the curves was large, which indicated that the failure mode of the joints during the stretching process may be similar to a certain degree. The damage started from the edge and gradually spreads to the middle. In summary, the following conclusions can be drawn: Araldite® 2012 adhesive can be used as a fast adhesive and, when the time was short, it had sufficient strength. Araldite® 2014 adhesive can be used as a durable adhesive to resist the erosion of the aging environment through its post-curing reaction.

### 3.5. Analysis of Failure Section Morphology

The fracture performance of the joints treated in an 80 °C/95%RH and 80 °C/pure water environment was analyzed through the surface macroscopic photos. Figure 12 shows the surface appearance of two adhesive joints after tensile fracture under two environments.

In Figure 12, the interface failure and tearing failure were separated by red lines. As a whole, the joints were basically all interface failures when they were not aged. Subsequently, the area of interface failure continued to decrease with the continuous increase of aging time, and it was converted to complete tear failure in 30 days. It can be clearly seen from Figure 12a that the Araldite® 2012 adhesive joint was completely interface failure when it was not aged. At 10 days, more than half of the failure area appeared as interface failure. At 20 days, the area of interface failure was reduced, and only nearly half of the failure area. At 30 days, there was no interface failure, which was manifested as a complete tear failure. In Figure 12b, the area of interface failure at 20 days was similar to that in Figure 12a, but at 20 days, there was only a small piece of interface failure at the center, and it appeared as complete tearing failure at 30 days. Therefore, it can be seen that the surface failure characteristics of bonded joints in Araldite® 2012 at 80 °C/pure water for 20 days and 30 days were very similar. That was the performance of the similarity of the average maximum failure load of the two kind of joints (3.46% difference). In Figure 12c,d, the failure trend of the joint section was similar to that of Figure 12a,b. In particular, in Figure 12d (Araldite® 2014 adhesive joint under 80 °C/pure water environment), the tearing areas of the 20-day failure surface and the 30-day failure surface were almost the same. The difference between the two average maximum failure loads was also not much (2.07%). Thus, the change of the failure surface was consistent with the trend of the average maximum failure load described above.

Consequently, combined with the post-curing phenomenon mentioned above, it can be seen that the reason for the gradual reduction of interface failure was that, as the aging time increased, the bonding force between the adhesive and the resin matrix continued to increase. The damage of the basalt fiber/epoxy resin interface in BFRP composites had been increasing, and the failure load of the joints had decreased, indicating that the BFRP composites were aging [36]. According to the conclusion of Sugimanet [37], owing to the thermal strain and expansion strain generated in the adhesive layer, the failure of the joints started at the end of the overlap, and then gradually spread to the central part of the joint as the load increased. That appearance was especially obvious in the 10-day failure situation in the figure. The overlapping ends produced compressive stress due to water absorption and expansion, and the tension near the center eventually caused the glue layer near the center to break.

In order to explore the deep-seated reasons why the damage of the basalt fiber/epoxy resin interface in the BFRP composite material increases with time, scanning electron microscopy analysis was performed on the joints with tear failure, as shown in Figure 13. Almost all the pictures in Figure 13 showed bare fibers. The red boxes are the broken fibers, and the yellow are the epoxy resin layers in the BFRP.

The exposed fiber in Figure 13 showed that the epoxy resin layer in the BFRP sheet was separated from the fiber mesh layer, which was the result of the continuous increase of the bonding force between the adhesive and the resin matrix [30]. Combined with the results of the DSC curves, it can be seen that, as the aging time increases, the glass transition temperature of the adhesive decreased, the glass transition temperature in the polymer chain decreased, the chain mobility increased, and the degree of crosslinking increased [29,35]. At the same time, the epoxy resin in the epoxy adhesive and the epoxy resin in the BFRP sheet had good compatibility. In this way, as time increased, the chain mobility increased; the greater the fluidity, the more significant the interaction between the epoxy adhesive and the surface resin layer chains. When these interactions were greater than the interactions between the fiber-resin layer inside BFRP, it will cause delamination and tearing [38].

## 4. Conclusions

In this paper, the various properties of Araldite® 2012 and Araldite® 2014 of BFRP-Al joints under two damp and heat conditions (80 °C/95%RH and 80 °C/pure water) were studied, and the following conclusions were obtained:Through the simulation of Fickian law on the experimental data, it was found that the saturated water absorption rate was largest in Araldite® 2012, followed by Araldite® 2014, and BFRP was the smallest. The diffusion coefficient did not increase with the increase of the saturated water absorption. The diffusion coefficient in Araldite® 2014 was the largest, followed by Araldite® 2012, and BFRP was the smallest.In the observation of the average maximum failure strength, it was found that Araldite® 2014 adhesive underwent a post-curing reaction at a high temperature of 80 °C, causing the joint strength to increase first and then decrease, indicating that it can resist the erosion of the aging environment through post-curing. Araldite® 2012 joint strength continued to decline, and the degree of aging continued to deepen, indicating that it was a fast adhesive.Through the observation of the load–displacement curve, it was found that the slope of the curve decreased with time, indicating that the stiffness of the bonded joint decreased with time. As a brittle adhesive, Araldite® 2014 had a bilinear curve, in which microcracks at the bilinear turning point would significantly affect the stiffness. Using macroscopic images to observe the failure interface of the joint, it was found that the surface failure transitions from the mixed failure in the early stage to the complete tear failure in the later stage, indicating that the BFRP material had been aged. In order to further explore the reasons for the increased tearing failure, microscopic SEM was used, and it found many basalt fiber filaments on the failure interface. This showed that the reason for the destruction of the basalt fiber/resin interface is the interaction between the epoxy resin in the adhesive and the epoxy resin in BFRP is greater than the force between the basalt fiber layer and the epoxy layer in the BFRP sheet.

## Figures and Tables

**Figure 1 polymers-13-02593-f001:**
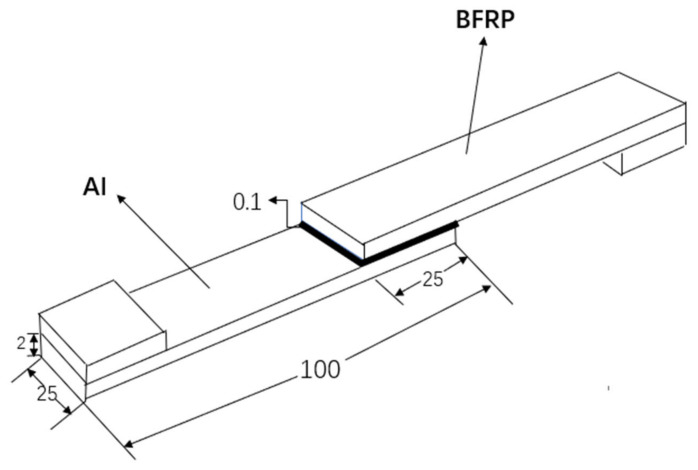
Configurational geometry and dimensions (mm).

**Figure 2 polymers-13-02593-f002:**
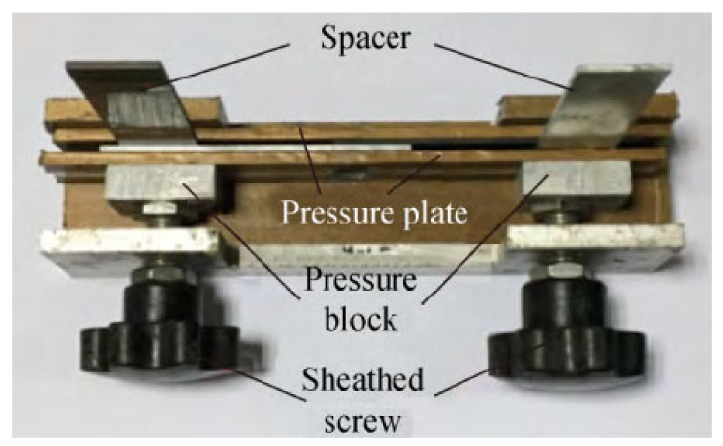
Adhesive fixture of single-lap joints.

**Figure 3 polymers-13-02593-f003:**
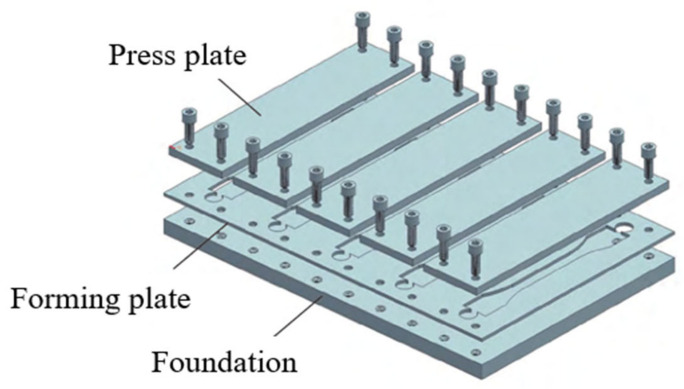
Standard model for dumbbell-shaped specimen.

**Figure 4 polymers-13-02593-f004:**
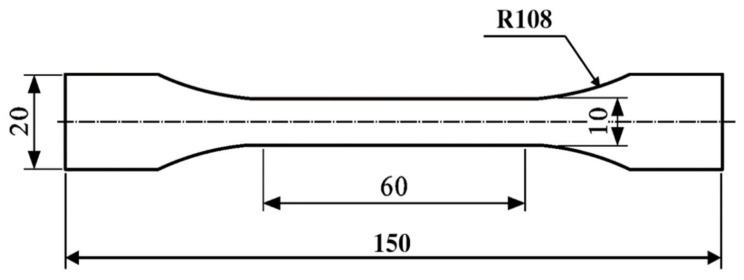
Dumbbell specimen geometry and dimensions in mm.

**Figure 5 polymers-13-02593-f005:**
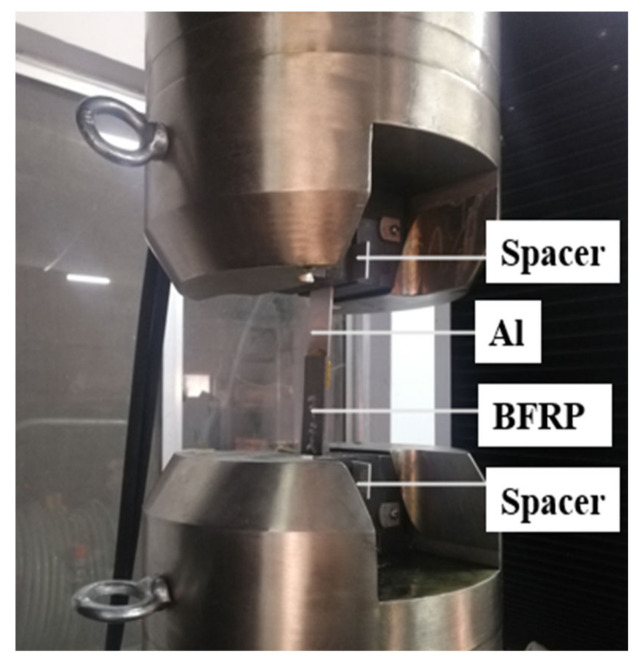
Quasi-static state tensile test site.

**Figure 6 polymers-13-02593-f006:**
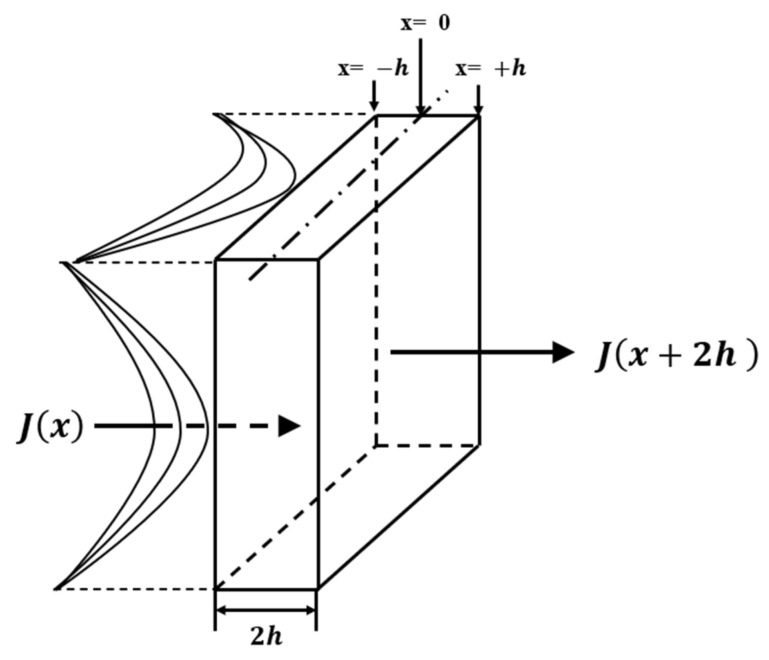
Diffusion in cuboid space.

**Figure 7 polymers-13-02593-f007:**
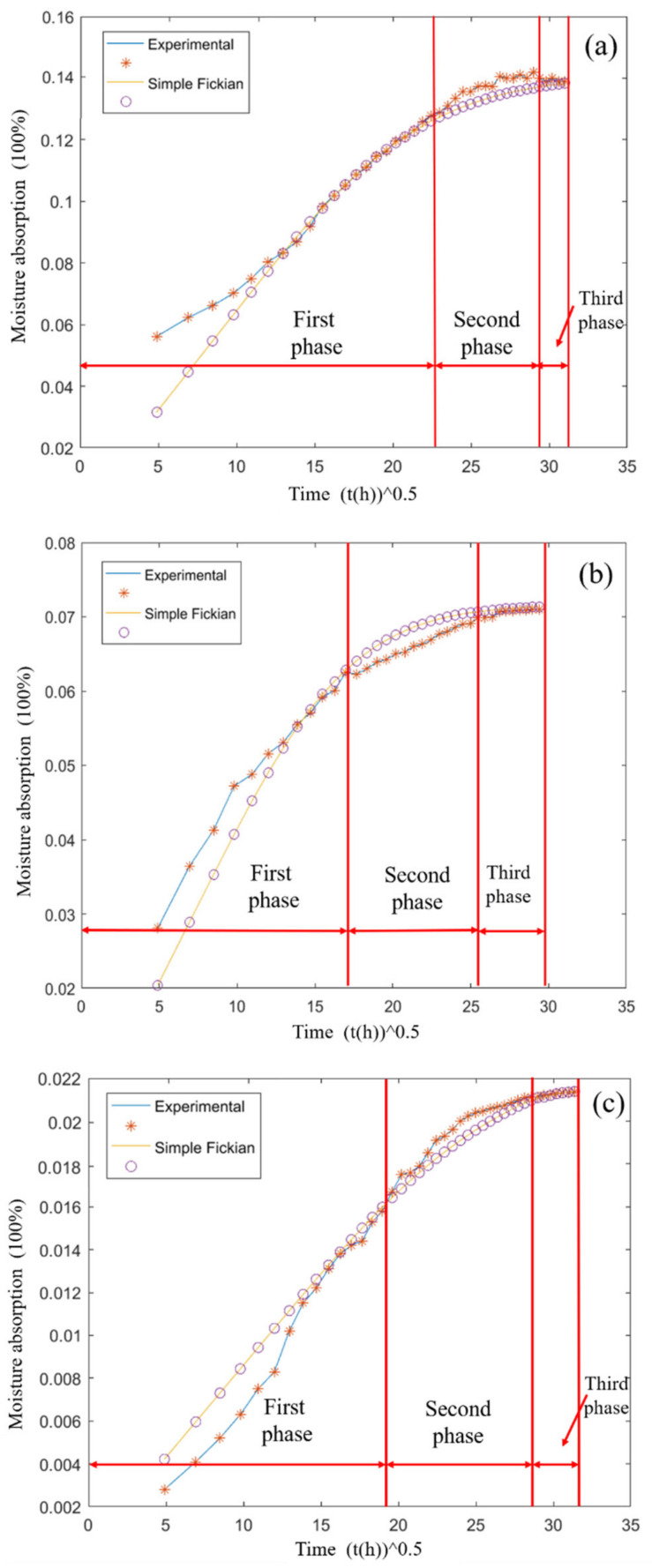
Moisture absorption law of three different materials: (**a**) the moisture absorption of Araldite® 2012 in 80 °C/pure water, (**b**) the moisture absorption of Araldite® 2014 in 80 °C/pure water, and (**c**) the moisture absorption of basalt fiber-reinforced polymer (BFRP) in 80 °C/pure water.

**Figure 8 polymers-13-02593-f008:**
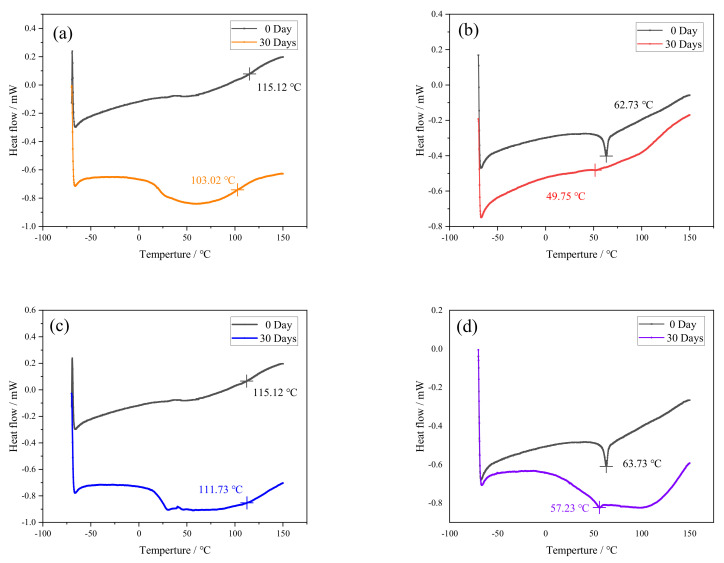
DSC thermogram of the adhesive: (**a**) Araldite® 2012 in 80 °C/95%RH, (**b**) Araldite® 2014 in 80 °C/95%RH, (**c**) Araldite® 2012 in 80 °C/pure water, and (**d**) Araldite® 2014 in 80 °C/pure water.

**Figure 9 polymers-13-02593-f009:**
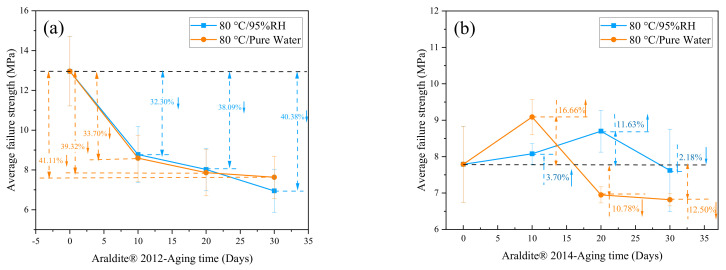
Failure strength of the adhesive joints in different environments: (**a**) Araldite® 2012 in 80 °C/95%RH and 80 °C/pure water, (**b**) Araldite® 2014 in 80 °C/95%RH and 80 °C/pure water.

**Figure 10 polymers-13-02593-f010:**
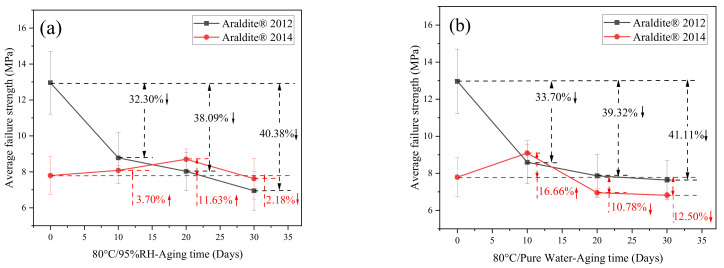
Failure strength of the adhesive joints at different environments: (**a**) Araldite® 2012 and Araldite® 2014 joints in 80 °C/95%RH and (**b**) Araldite® 2012 and Araldite® 2014 joints 80 °C/pure water.

**Figure 11 polymers-13-02593-f011:**
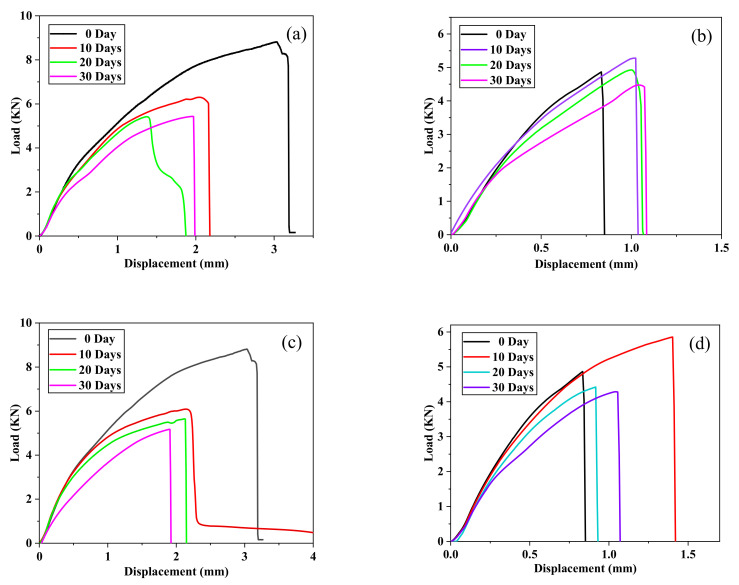
Load–displacement curves: (**a**) Araldite® 2012 joints in 80 °C/95%RH, (**b**) Araldite® 2014 joints in 80 °C/95%RH, (**c**) Araldite® 2012 joints in 80 °C/pure water, and (**d**) Araldite® 2014 joints in 80 °C/pure water.

**Figure 12 polymers-13-02593-f012:**
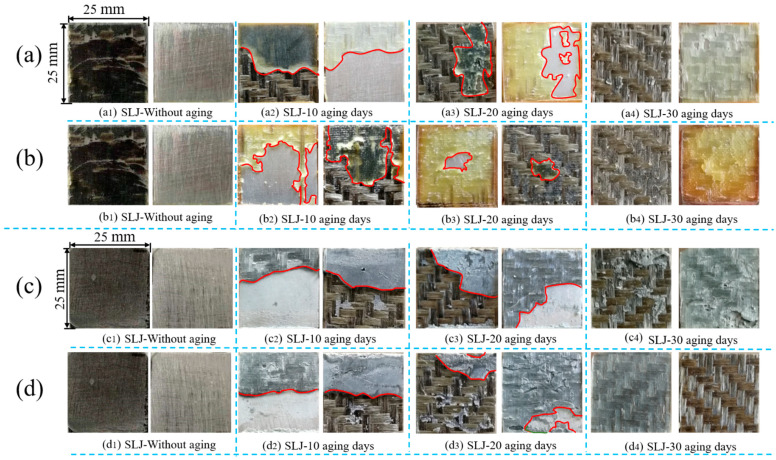
The surface of different adhesive joints in different environments: (**a**) Araldite® 2012 joints in 80 °C/95%RH, (**b**) Araldite® 2012 joints in 80 °C/pure water, (**c**) Araldite® 2014 joints in 80 °C/95%RH, and (**d**) Araldite® 2014 joints in 80 °C/pure water.

**Figure 13 polymers-13-02593-f013:**
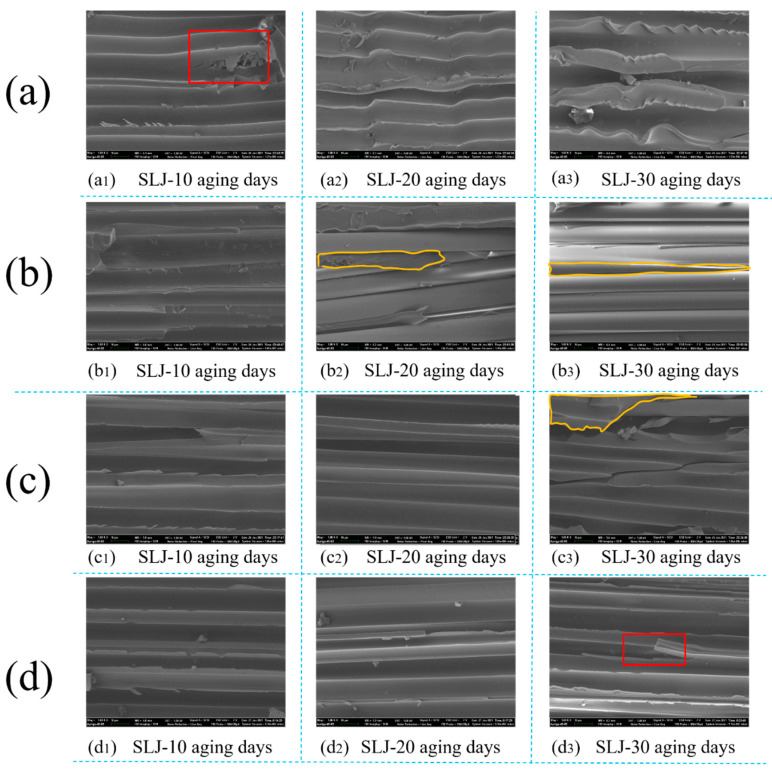
The SEM micrograph of different adhesive joints in different environments: (**a**) Araldite® 2012 joints in 80 °C/95%RH, (**b**) Araldite® 2012 joints 80 °C/pure water, (**c**) Araldite® 2014 joints in 80 °C/95%RH, and (**d**) Araldite® 2014 joints 80 °C/pure water.

**Table 1 polymers-13-02593-t001:** Material properties of basalt fiber unidirectional fabric.

Surface Density/(g/cm^2^)	Tensile Strength/MPa	Young’s Modulus/(MPa)	Nominal Thickness/mm	Single Fiber Size/μm
600	2100	105	0.115	11

**Table 2 polymers-13-02593-t002:** 5052 aluminum alloy material properties.

Density/(kg·m^3^)	2730
Young’s modulus/(MPa)	70,000
Poisson’s radio	0.33
Yield strength/MPa	227
Tensile strength/MPa	378

**Table 3 polymers-13-02593-t003:** Araldite® 2012 and 2014 parameters.

	Araldite® 2012	Araldite® 2014
Young’s modulus, E (GPa)	1.65	4.36
Shear modulus, G (GPa)	0.25	1.2
Density/(kg·m^3^)	1.18	1.6
Poisson’s radio	0.43	0.33

**Table 4 polymers-13-02593-t004:** Moisture uptake of three different materials.

Samples (Under Pure Water)	Saturation Moisture Uptake Mm (%)	Diffusion Coefficient D × 10^−3^ (mm^2^/s)	Thickness (T = 2*h*, mm)
Araldite® 2012	14.05	1.78	2
Araldite® 2014	7.15	2.67	2
BFRP-pure water	2.15	1.22	2

## Data Availability

Data sharing not applicable.

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
