# Peer review of "Comparative Failure Study of Different Bonded Basalt Fiber-Reinforced Polymer (BFRP)-AL Joints in a Humid and Hot Environment"

_polymers, 2021, doi:10.3390/polym13162593_

Round 1

Reviewer 1 Report

Dear colleagues,

The article entitled “Comparative Failure Study of Different Bonded BFRP-AL Joints in Humid and Hot Environment” has the main goal of evaluating the influence of high temperature and high humidity on the bonding structure between basalt fiber reinforced resin composite materials BFRP and Al alloy, using Araldite® 2012 and Araldite® 2014 to make single lap joints. Article is interesting and well written, apart from some grammar/distraction mistakes. Please revise them. Intended application could be further detailed. Also, I am not sure of your novelty. Please compare your work to the ones presented in publications with DOI: 10.1016/j.compstruct.2021.114013, 10.1080/00218464.2020.1712656 and 10.1016/j.ijadhadh.2018.05.027, and reinforce it.

Abstract: Novelty and impact in society is unclear.

Introduction:

Line 57: please introduce the abbreviation SLJ

Author Response

Thank you very much for affirmation of our work. The reviewer's comments are of great help to our current manuscript. The point to point responds to the reviewer’s comments are listed as following.

1.The article entitled “Comparative Failure Study of Different Bonded BFRP-AL Joints in Humid and Hot Environment” has the main goal of evaluating the influence of high temperature and high humidity on the bonding structure between basalt fiber reinforced resin composite materials BFRP and Al alloy, using Araldite® 2012 and Araldite® 2014 to make single lap joints.

(1) Article is interesting and well written, apart from some grammar/distraction mistakes. Please revise them.

Response: Thank you very much. I am so sorry for the mistakes in grammar/distraction. We have been revised them in line 20-22, 129-133, 175-177, 229-231.

(2) Intended application could be further detailed.

Response: Thank you. Intended application part has been elaborated in the abstract and introduction. Please look at line 9-10 and 35-39.

(3) Also, I am not sure of your novelty. Please compare your work to the ones presented in publications with DOI: 10.1016/j.compstruct.2021.114013, 10.1080/00218464.2020.1712656 and 10.1016/j.ijadhadh.2018.05.027, and reinforce it.

Response: Thank you. We have read three publications carefully, these three publications have a certain inspiration for our paper, also have some similarity with us. However, these three publications all use one adhesive called Araldite®2015, this paper uses comparatively two adhesives which are tough adhesive Araldite®2012 and brittle Araldite® 2014. Both of these adhesives are commonly used in the automotive industry. By comparing and studying their performance on BFRP-Al joints, it is of great value to guide their application in the automotive industry. In short, our innovation lies in the comparative study of two adhesives with different properties.

2、Introduction:

Line 57: please introduce the abbreviation SLJ

Response: Thank you very much. Your observation is so careful. We have been introduced it. Now it is in line 61-62.

Reviewer 2 Report

Please define all acronyms before their first appearance, i.e. BFRP, Al ..

Line 19 “it is increased .. “may is wrong ..please check

The “ Load-Displacement curve” is not an indicator for assessing the bonding resistance. Either calculation G is much better

The fist 1. And second Conclusion it is repeated in abstract ! this is not necessary

Therefore I suggest to avoid repeating the abstract in conclusion, so you have to make two distinct section which do not overlap.  

Justification of using this adhesive “Araldite” is very vague as not clear if this is used at industrial scale and not

This phrase “Test the changes in the mechanical…” is to long and do not make sense ..please reformulate it

Overall the English requires major improvement in overall manuscript.

The scientific novelty is rather limited in this work

Why was selected Single lap joint and not DCB which is more common type of test of adhesive bonding ?

How easy can be reproduced this process at industrial scale “The surface treatment of 5052 Al substrate was firstly polished with 80# sandpaper to remove the oxide layer”?

Please provide a ref for this standard “NF ISO 527-2-2012”

Please check this because for sure the sample failed and not the machine “The testing machine was stretched at a constant speed of 2mm/min until it fails.”

Provide a reference for “Fickian Second Law”

It is interesting to provide details of “In Dhakal [16] et al. mentioned 316 three mechanisms of water..” however not clear how was measured these 3 , or the last one here in this paper otherwise looks rather a speculation

In text you said 20 days and in Figure 8 you shows 30days !

Author Response

Thank you very much for affirmation of our work. The reviewer's comments are of great help to our current manuscript. The point to point responds to the reviewer’s comments are listed as following.

  1. Please define all acronyms before their first appearance, i.e. BFRP, Al .

       Response: Thank you very much. We have defined all acronyms in the           location of their first appearance.

  1. Line 19 “it is increased .. “may is wrong ..please check

        Response: Thank you. We have revised it which is in line 20-22 and page 1    now.

  1. The “ Load-Displacement curve” is not an indicator for assessing the bonding resistance. Either calculation G is much better

        Response: Thank you. Your proposal is very useful and we will use it in  future research.

  1. The fist 1. And second Conclusion it is repeated in abstract ! this is not necessary

        Response: Thank you. Your suggestion is very pertinent. We have modified in the conclusion to make these two parts different. (line 544-566,page 22)

  1. Justification of using this adhesive “Araldite” is very vague as not clear if this is used at industrial scale and not.

        Response: Thank you. The adhesive selected in this article is currently widely used in the automotive industry (We have revised and emphasized in the introduction, line 35-39,page 1), and also widely studied in academia, such as:

Wang S, Wang S, Li G, et al. Dynamic response and fracture analysis of basalt fiber reinforced plastics and aluminum alloys adhesive joints[J].  Composite Structures, 2021, 268(3):114013.

Na J, Mu W, Qin G, et al. Effect of temperature on the mechanical properties of adhesively bonded basalt FRP-aluminum alloy joints in the automotive industry[J]. International Journal of Adhesion & Adhesives, 2018, 85:138-148.

  1. This phrase “Test the changes in the mechanical…” is to long and do not make sense ..please reformulate it.

        Response: Thank you. We are so sorry for our mistake. And it has been revised in line 131-136 and page 3.

  1. The scientific novelty is rather limited in this work.

        Response: Thank you. This paper uses comparatively two adhesives which are tough adhesive Araldite®2012 and brittle Araldite® 2014. Both of these adhesives are commonly used in the automotive industry. By comparing and studying their performance on BFRP-Al joints, it is of great value to guide their application in the automotive industry. In short, our paper focuses on the comparative study of two adhesives with different properties.

  1. Why was selected Single lap joint and not DCB which is more common type of test of adhesive bonding?

        Response: Thank you. The authors refer to most studies and found that they all study Single Lap joints. Single Lap joints are easy to make, simple to test, and economical.

  1. How easy can be reproduced this process at industrial scale “The surface treatment of 5052 Al substrate was firstly polished with 80# sandpaper to remove the oxide layer”?

        Response: Thank you. It is easy.

  1. Please provide a ref for this standard “NF ISO 527-2-2012”.

        Response: Thank you. It has been revised. Please look at line 200, page 6 and citation [24].

  1. Please check this because for sure the sample failed and not the machine “The testing machine was stretched at a constant speed of 2mm/min until it fails.”

       Response: Thank you for your careful work. The error has been modified in line 234-235 in page 7.

  1. Provide a reference for “Fickian Second Law”.

        Response: Thank you. The reference has been provided in line 242, page 7 and citation [25].

  1. It is interesting to provide details of “In Dhakal [16] et al. mentioned 316 three mechanisms of water.” however not clear how was measured these 3, or the last one here in this paper otherwise looks rather a speculation.

        Response: Thank you. This is the conclusion put forward by Dhakal et al., and considering the properties of basalt fiber and saturated water absorption and diffusion coefficient of BFRP sheet in this paper, we put forward the conclusion that the third mechanism should not exist or rarely exist after the three mechanisms. (line 334-337, page11)

  1. In text you said 20 days and in Figure 8 you shows 30 days !

        Response: Thank you for your careful work. The wrong has been modified. (line 350,page12)

Round 2

Reviewer 2 Report

.